# Quantum sensing and metrology with free electrons

Cruz I. Velasco[1] & F. Javier García de Abajo [1,2] ✉

The quantum properties of matter and radiation can be leveraged to surpass classical limits of sensing and detection. Quantum optics does so by creating and measuring nonclassical light. However, better performance requires higher photon-number states, which are challenging to generate and detect. Here, we combine photons and free electrons to solve the problem of generating and detecting high-number states well beyond those reachable with light alone and further show that an unprecedented level of sensitivity and resolution is gained based on the measurement of free-electron currents after suitably designed electron–light interactions. Our enabling ingredient is the strong electron–light coupling produced by aloof electron reflection on an optical waveguide, leading to the emission or absorption of a high number of guided photons by every single electron. We establish through rigorous theory that, by combining electron-beam splitters with two electron–waveguide interactions, the sensitivity to detect optical-phase changes can be enhanced dramatically using currently attainable technology. These results inaugurate a disruptive quantum technology relying on free electrons and their strong interaction with waveguided light.

Quantum sensing and metrology encompass a wide range of techniques that exploit the quantum properties of microscopic systems, such as entanglement and wave-like behavior, to achieve improved sensitivity and resolution compared to classical methods[1,2]. Quantum optics provides a powerful platform for implementing these ideas, where the generation and detection of nonclassical states of light have enabled measurements with superior sensitivity. For example, squeezed states played a key role in detecting gravitational waves, where a light interferometer was used to sense spatial modulations 18 orders of magnitude smaller than the wavelength of the lasers used[3]. More exotic states of light are also helpful in this context, such as the so-called NOON states, $[|N, 0\rangle + |0, N\rangle]/\sqrt{2}$, where the first and second entries in each ket denote the numbers of quanta in two different channels (e.g., photons in different waveguides). NOON states represent maximally entangled superpositions of $N$-photon states that saturate the Heisenberg uncertainty limit[4], enabling measurements with both super-sensitivity and super-resolution by beating the shot-noise and diffraction limits, respectively. At high $N$, these properties

make NOON states powerful tools in quantum sensing and metrology[4,5], quantum lithography[6,7], and even quantum computing[8]. However, high-$N$ NOON states (commonly referred to as high-NOON states) remain extremely challenging (e.g., $N = 5$ generation is currently only possible at sub-Hz rates[9], and $N = 10$ at mHz[10]).

In a separate effort, free electrons have recently emerged as powerful carriers of quantum information, which is encoded in their energy and propagation direction via states that can be prepared through interaction with both classical[11] and quantum[12] light using ultrafast electron microscopy techniques[13–21]. In addition, free electrons exhibit wave-like behavior that manifests in diffraction, interference, and superposition. These attributes have been exploited for the electron-based generation of quantum light. For example, following interaction with an optical cavity and electron post-selection, energy loss events experienced by individual electrons can herald the generation of quantum states of light such as single[22,23] or multiple[24,25] photon-number states. In addition, protocols have been theoretically proposed for the free-electron-based generation of

[1]ICFO-Institut de Ciencies Fotoniques, The Barcelona Institute of Science and Technology, Barcelona, Spain. [2]ICREA-Institució Catalana de Recerca i Estudis Avançats, Barcelona, Spain. ✉e-mail: javier.garciadeabajo@nanophotonics.es

squeezed[26], Gottesman–Kitaev–Preskill[27], and entangled-pair[28] photon states. Quantum entanglement between electrons and optical excitations has recently been demonstrated in two separate experiments[29,30], while quantum tomography has been applied to free electrons[16], photoemitted electrons[31], and optical cavity modes probed by free electrons[32].

These emerging electron-based quantum technologies rely on the ability to couple electrons and light efficiently. However, the electron–photon interaction is generally low, and, for example, the number of photons created by a single electron in a single mode has remained below unity in experiments. To address this problem, several schemes have been proposed to increase such interaction[33–36], and in particular, phase-matched coupling of electrons moving parallel and outside a waveguide[22] constitutes a promising direction that has been argued to lead to the generation of high photon-number states[36]. The combination of electron state superposition and strong electron–photon coupling can be anticipated to facilitate the generation of high-NOON states as well as the implementation of quantum sensing and metrology protocols that may potentially exceed the capabilities of all-optical schemes.

Here, we introduce free electrons as quantum probes on par with photons, enabling advanced quantum sensing and metrology protocols that extend the field well beyond its present frontiers. Specifically, we propose a class of systems that integrate free-electron wave optics with the interaction between free electrons and waveguided photonic modes. A central component in this approach is the design of realistic electron–light couplers capable of generating multiple photons per electron within a single photonic mode without introducing electron decoherence. By leveraging this strong electron–light interaction and preparing electrons in coherent superpositions of different paths, we establish through rigorous theory protocols that enable both super-sensitivity and super-resolution in the measurement of optical phases via the detection of free-electron currents alone. These results emphasize the potential of free electrons as a powerful platform for quantum sensing and metrology, enabling previously unattainable capabilities through the integration of existing technologies.

We remark that our approach to free-electron sensing and metrology is based on a large electron–light interaction strength, for which we propose an innovative scheme: a glancing electron trajectory assisted by a DC field (see Fig. 1a). This configuration serves two purposes:

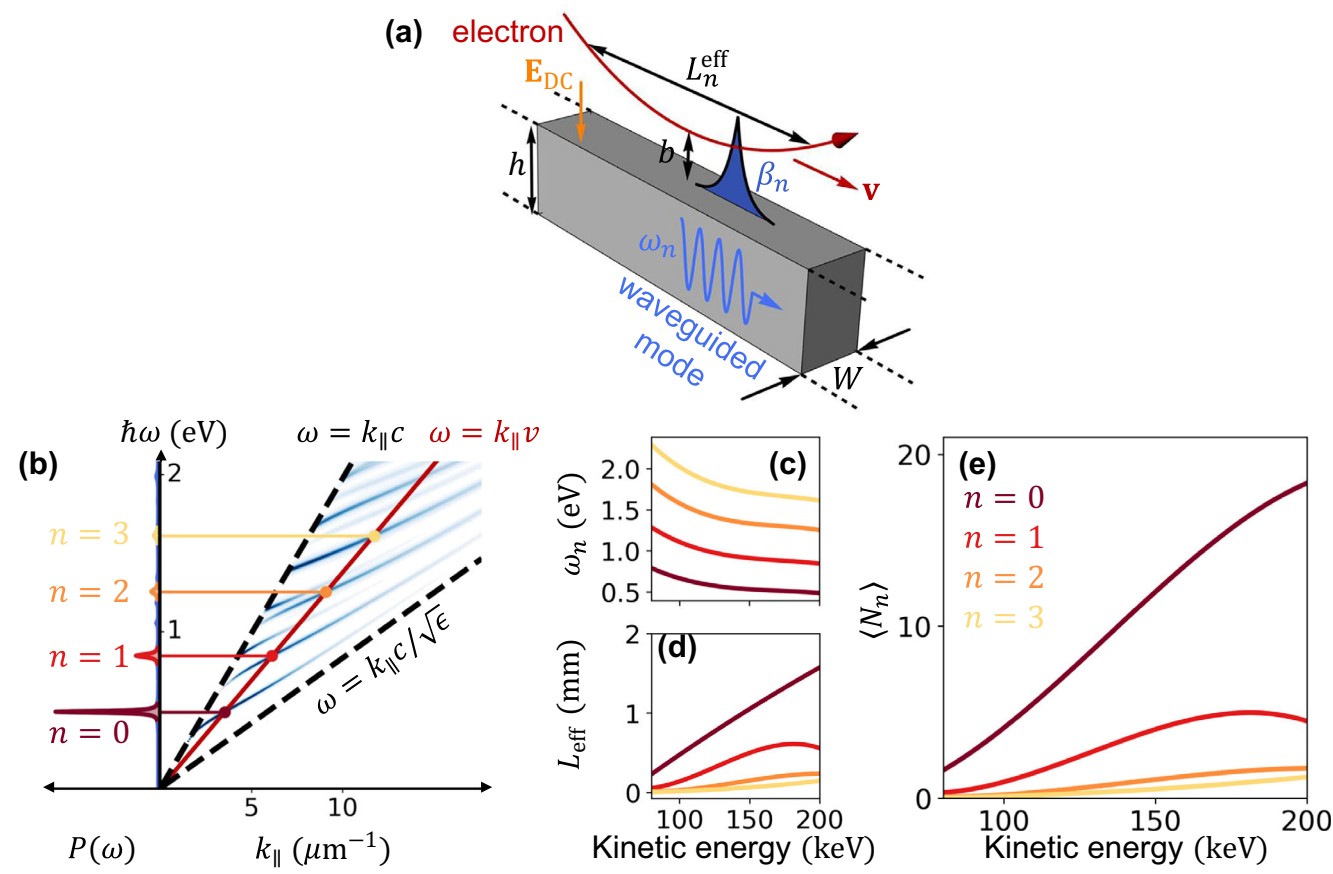

**Fig. 1 | Multiple photon generation by individual free electrons. a** Sketch of an electron reflected at a grazing angle from an optical dielectric waveguide (rectangular cross section, width $W$, height $h$), following a bent trajectory with parallel velocity $\mathbf{v}$, a minimum distance $b$ from the waveguide, and a mode-dependent effective interaction length $L_n^{\mathrm{eff}}$. The electron is repelled from the surface by a normal electric DC field $\mathbf{E}_{\mathrm{DC}}$. Waveguide modes with frequencies $\omega_n$ are excited with amplitudes $\beta_n \propto L_n^{\mathrm{eff}}$, provided they fulfill the phase-matching condition discussed in (**b**). **b** Mode dispersion for a diamond waveguide ($W = 600$ nm, $h = 800$ nm) as a function of photon wave vector $k_\parallel$ and frequency $\omega$, visualized through the electron energy-loss probability (blue), revealing modes confined between the vacuum and bulk-diamond light lines (dashed lines) for photon energies below the material band gap (5.5 eV). The phase-matching condition $\omega = k_\parallel v$ is indicated in red for a 200 keV electron ($v \approx 0.7\,c$), together with the loss probability $P(\omega)$ (left curves, arbitrary units). **c**–**e** Frequency $\omega_n$ (**c**), effective interaction length $L_n^{\mathrm{eff}}$ (**d**), and average number of generated photons $\langle N_n \rangle$ (**e**) for the four dominant photonic modes [$n = 1 - 4$, color-matched with the probability peaks in (**b**); see legend in (**e**)], plotted as functions of electron kinetic energy for $b = 200$ nm and $E_{\mathrm{DC}} = 25$ V/mm.

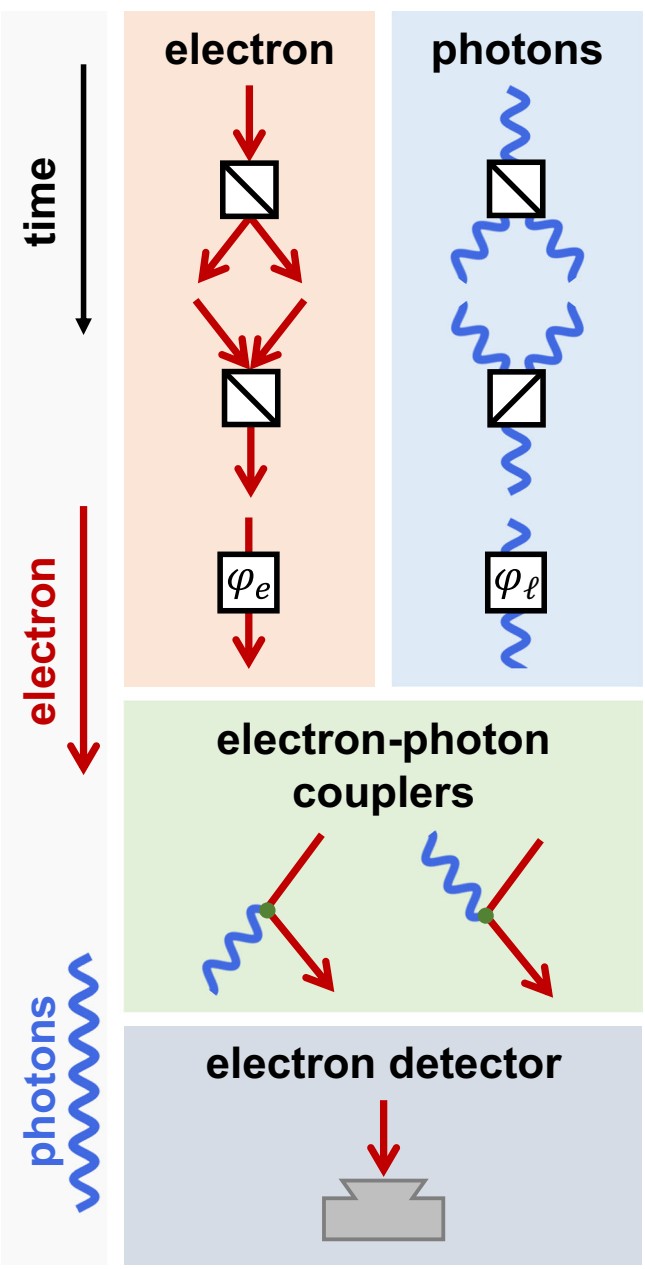

**Fig. 2 | Building blocks of quantum free electronics.** QUAFE components include beam splitters/mixers and phase shifters for electrons (phase $\varphi_e$) and waveguided photons ($\varphi_\ell$), as well as electron-photon couplers (for light emission/absorption; left/right) and electron detectors. Time flows downward in each diagram. Electrons and photons are represented by red arrows and blue waves, respectively.

(1) *Introducing a minimum transverse distance (perpendicular to the beam) of electron–waveguide interaction.* The electron is deflected at a distance $b \sim 200$ nm away from the surface of the waveguide to prevent inelastic transitions (e.g., electron-hole pairs). The latter decay fast outside the waveguide material and, thus, do not couple to the electron beyond $b$. In contrast, waveguided modes extend farther away from the surface and are still efficiently excited.

(2) *Increasing the effective electron–waveguide interaction distance $L^{\mathrm{eff}}$.* This ensures the generation of a large number of photons per electron. For a realistic choice of material and electron parameters, our calculations below yield $L^{\mathrm{eff}} \sim 100$'s μm, for which an average of tens of waveguided photons is generated per electron, with lossless propagation

distances of centimeters in state-of-the-art integrated photonic waveguides.

In addition, this scheme should be insensitive to surface roughness and imperfections below a few nanometers.

In what follows, we present a detailed description of the interaction scheme and first apply it to the generation of high NOON states for tutorial purposes. We then construct metrology schemes based on electron detection alone, which we emphasize as a practical approach to realizing a similar level of sensitivity and resolution as obtained with NOON states comprising $N \sim$ tens photons, generated at a rate of $\sim 10^9$ per second as dictated by the electron current. An important additional element in our proposal involves the lateral splitting and mixing of electron beams (e-beams), as experimentally demonstrated using transmission gratings[37,38]. Our schemes are based on modular elements involving electron splitting/mixing and efficient electron–light coupling, as discussed in Fig. 2. Based on these tools, the presented scheme goes well beyond the current technological capabilities of photonics-based sensing and metrology. We remark that this is possible thanks to the ability to generate tens of photons per electron, well beyond current experimental realizations of photon-number-state generation heralded by electron energy losses, which still lie at a level below one photon per electron[22–25,35,36,39].

## Results and discussion

### Efficient electron–photon coupler

The key element enabling the present proposal for free-electron-based quantum sensing and metrology is the efficient generation and detection of quantum states of light through electron–photon interaction. To achieve this, we consider the configuration sketched in Fig. 1a, where an energetic free electron is incident at a grazing angle and reflects from a one-dimensional waveguide with rectangular cross-section. We note that other waveguide geometries may also be suitable, provided the guide-mode fields extend beyond the material boundary, allowing the electron to couple to them. A repulsive DC electric field, oriented normal to the waveguide surface, is applied to mitigate the effects of close collisions between the electron and the material, which are detrimental to electron coherence (see Supp. Fig. S1). The minimum electron–surface distance $b$, determined by the incident glancing angle ($\theta_0 \lesssim 1$ mrad), the electron kinetic energy, and the electric field amplitude $E_{\mathrm{DC}}$, is chosen such that inelastic excitations above the band gap of the material (e.g., electron-hole pairs) are negligible, while coupling to guided modes remains sizeable. This is feasible because the fields of waveguided modes penetrate further into the vacuum, whereas those associated with high-energy inelastic excitations decay more rapidly with distance to the surface.

Given the small electron velocity along the perpendicular direction, we analyze the electron–photon coupling by considering parallel trajectories, in which the electron experiences an energy-loss probability $d\Gamma(x, \omega)/dz$ (i.e., the generation of a photon of frequency $\omega$ is associated with an electron energy loss $\hbar\omega$). This quantity depends on the electron–surface distance $x$ and is normalized per unit length along the waveguide direction $z$. The probability is resolved in specific energy losses $\hbar\omega$, and we normalize it such that $\int_0^\infty d\omega \, [d\Gamma(x,\omega)/dz] = dP(x)/dz$ gives the total inelastic scattering probability per unit electron path length. We compute the electron energy-loss probability $d\Gamma(x, \omega)/dz$ (identical to the photon-generation probability per unit path length within the spectral region in the band gap of the material) from the self-induced field acting back on the electron[40] using the boundary-element method[41] (BEM) as an efficient numerical approach.

The electron maintains an approximately constant velocity $\mathbf{v}$ parallel to the waveguide (nonrecoil approximation), imposing a phase-matching condition $\omega = k_\parallel v$ between the frequency $\omega$ and the parallel wave vector $k_\parallel$ of the excitations that can be created in the

waveguide[22]. By varying the electron velocity to scan a wide ($k_\parallel$, $\omega$) range and plotting the electron energy-loss probability, we reconstruct the dispersion diagram shown in Fig. 1b for a diamond waveguide. Here, waveguide modes (blue features) appear within the region flanked by the vacuum and bulk-diamond light lines (dashed). For illustration, the phase-matching condition is indicated for 200 keV electrons (red line), along with the corresponding loss probability (left curves). Guided modes are well-defined at energies within the band gap of the material, where they exhibit zero linewidth. For visualization, we introduce a small imaginary component Im{$\epsilon$} in the permittivity within the band gap, which broadens the modes into finite-width features. We define the photon-generation probability per unit length as $\int_n d\omega \, [d\Gamma(x, \omega)/dz] = dP_n(x)/dz$ for each waveguide mode $n$, where the frequency integral spans the corresponding peak in the energy-loss spectrum. This integral is approximately independent of Im{$\epsilon$} as long as this quantity remains small compared with |$\epsilon$|. Finally, we note that the deviation from the perfectly parallel trajectory in the glancing geometry leads to additional mode broadening, which we neglect in the following analysis.

We chose diamond (permittivity $\epsilon \approx 5.8$, see ref. 42) as a waveguide material because its large band gap of 5.5 eV allows for nearly lossless propagation over tens of millimeters for modes with energies below this value. Alternative materials, such as silicon and germanium, could also be employed, as their relatively low losses can enable even longer propagation distances on the order of hundreds of millimeters for modes with energies below their band gaps of 1.12 eV and 0.67 eV, respectively. However, their smaller band gaps would necessitate adjusting the waveguide dimensions to ensure that the populated mode frequencies remain well below these limits.

The crossing points between the electron line $\omega = k_\parallel v$ and the guided modes determine the frequencies at which they are excited as a function of electron energy (Fig. 1c). Upon numerical inspection, the photon-generation probability per unit path length exhibits an exponential decay $dP_n(x)/dz \propto e^{-2x/\lambda_{\perp n}}$ as the electron moves away from the waveguide, consistent with the evanescent nature of the mode field outside the material. Based on the total light wave vector in vacuum ($\omega/c$), we anticipate the decay length to be approximately $\lambda_{\perp n} \approx (k_\parallel^2 - \omega^2/c^2)^{-1/2}$, which increases for modes closer to the vacuum light line. This result is corroborated by fitting the BEM-calculated electron energy-loss probability as a function of distance $x$. In what follows, we fix the waveguide width and height to $W = 600$ nm and $h = 800$ nm, respectively, yielding a decay length $\lambda_{\perp 0} \approx 290$ nm for the $n = 0$ mode with 200 keV electrons.

For simplicity, we consider a uniform repulsive field established between the waveguide (assumed to be slightly doped to act as a gate) and a nearby gate in the region near the waveguide (see Supp. Fig. S2 for a possible implementation). In addition to $E_{DC}$, the electron trajectory is dependent on the electron-surface distance $x_0$ and the angle $\theta_0$ between the electron velocity vector and the surface plane at the point of entrance in the capacitor region. We take $x_0 > 2\lambda_{\perp n}$, such that the electron-mode coupling occurs near the bending point of the trajectory. Here, we fix the minimum electron–waveguide distance to $b = 200$ nm, and set $x_0 = 500$ nm and $E_{DC} = 25$ V/mm. The value of $b$ determines the incidence angle for each electron energy (e.g., $\theta_0 = 0.2$ mrad at 200 keV). We also include the attractive image electron–waveguide interaction energy (see detailed calculation in Supp. Secs. S1 and S2), which plays a small but discernible role in the electron trajectory (see Methods).

The total photon-generation probability for each mode $n$ is obtained by integrating $dP_n(x)/dz$ over the parabolic electron trajectory $x = x_e(z)$, yielding $P_n = \int dz \, dP_n[x_e(z)]/dz$. This allows us to define a mode-dependent effective interaction length $L_n^{\text{eff}}$, such that $P_n = L_n^{\text{eff}}[dP_n(x = 0)/dz]$. After some algebra, we can express $L_n^{\text{eff}} = e^{-2b/\lambda_{\perp n}} \sqrt{\pi m_e \gamma v^2 \lambda_{\perp n}/eE_{DC}} \, g_{\text{im}}$ in terms of $b$, $E_{DC}$, and $v$, where $m_e$

and $-e$ are the electron mass and charge, $\gamma = 1/\sqrt{1 - v^2/c^2}$, and $g_{\text{im}}$ is an order-unity factor accounting for the image attraction between the electron and the waveguide (see Methods and Supp. Secs. S1–S3).

For the parameters under consideration, we find $L_n^{\text{eff}}$ in the millimeter range (Fig. 1d). Lower-order modes exhibit substantially smaller $L_n^{\text{eff}}$ values, as they lie closer to the light line in the dispersion diagram (see Fig. 1a). In addition, $L_n^{\text{eff}}$ increases with electron energy, since phase-matching occurs increasingly closer to the light line.

Starting from initially unpopulated waveguide modes, a photon-generation probability $P_n$ must be interpreted as a Poissian distribution of number-state populations in mode $n$[12,43,44] with an average number of excited photons given by $\langle N_n \rangle = P_n = |\beta_n|^2$ (see below), where $\beta_n$ is the coupling amplitude[12]

$$\beta_n = \frac{e}{\hbar\omega_n} \int_{-\infty}^{\infty} dz \, \hat{\mathbf{z}} \cdot \mathbf{E}_n(z) \, e^{-i\omega_n z/v}, \tag{1}$$

which is proportional to the spatial Fourier transform of the normalized electric field $\mathbf{E}_n$ associated with mode $n$ along the $\mathbf{v} = v\hat{\mathbf{z}}$ direction. With the parameters above, we find relatively high values of $\langle N_n \rangle$ (see Fig. 1e), reaching up to $\langle N_0 \rangle \sim 18$ photons in the lowest-order waveguide mode for 200 keV electrons, where phase-matching occurs at a photon energy of 0.48 eV.

From a practical perspective, this scheme requires a focused e-beam that remains well collimated over a propagation distance of several hundred microns or even a few millimeters. We model the beam as Gaussian, with its focal plane positioned at the point of closest approach to the waveguide. The focusing and collimation are characterized by the waist $w_0 = \lambda_e/\pi\Delta\theta$ and the Rayleigh range $z_R = \lambda_e/\pi(\Delta\theta)^2$, where $\Delta\theta$ is the beam divergence angle around a central angle $\theta_0$, while $\lambda_e$ is the electron de Broglie wavelength. For 200 keV electrons and $\Delta\theta \approx 0.01$ mrad, as achievable in state-of-the-art electron microscopes, we obtain $\lambda_e \approx 2.5$ pm, $z_R \approx 8$ mm, and $w_0 \approx 80$ nm. We anticipate that the effect of having different electron–waveguide closest distances when varying $\Delta\theta$ could be compensated by increasing $E_{DC}$ along the waveguide, such that the less glancing trajectories reaching the waveguide further along the waveguide experience a larger DC field that compensates for the higher normal kinetic energy (see Supp. Fig. S3), thus producing a uniform coupling to waveguide modes. In addition, the transverse width of the e-beam should be negligible compared to the lateral extension of the waveguide to ensure a uniform photon-generation probability.

## Quantum description of the electron–waveguide interaction

For a highly collimated e-beam with a small lateral extent compared to the spatial variation scale of the optical fields to which it is exposed, the electron wave function can be factorized as $\psi_\parallel^e \psi_\perp^e$ (i.e., a product of components parallel and perpendicular to the electron velocity vector $\mathbf{v}$). The details of $\psi_\perp^e$ are not relevant to our analysis, and therefore, we disregard them, except for the consideration of different electron paths.

We assume that the waveguide and the electron form a closed, lossless system, with energy transferred between the longitudinal motion of the electron and the guided modes, such that inelastic losses (e.g., due to electron-hole pair generation) are negligible. It is convenient to define creation and annihilation operators $a_n^\dagger$ and $a_n$ for the optical waveguide modes indexed by $n$. The photonic subsystem is then described in terms of photon-number states $|\{N_n\}\rangle$, where $N_n$ denotes the photon number in mode $n$ of frequency $\omega_n$. We also assume that the incident electron is prepared with a well-defined kinetic energy $\mathcal{E}_0$ (small broadening compared with the photon energies). From the assumption of energy conservation and the absence of losses, the electron energy changes to $\mathcal{E}_0 - \hbar\sum_n N_n\omega_n$ upon the creation of a photon-number state $|\{N_n\}\rangle$. The electron state is thus uniquely defined once we know the photonic state, and therefore, we can represent the combined electron–light system just by $|\{N_n\}\rangle$.

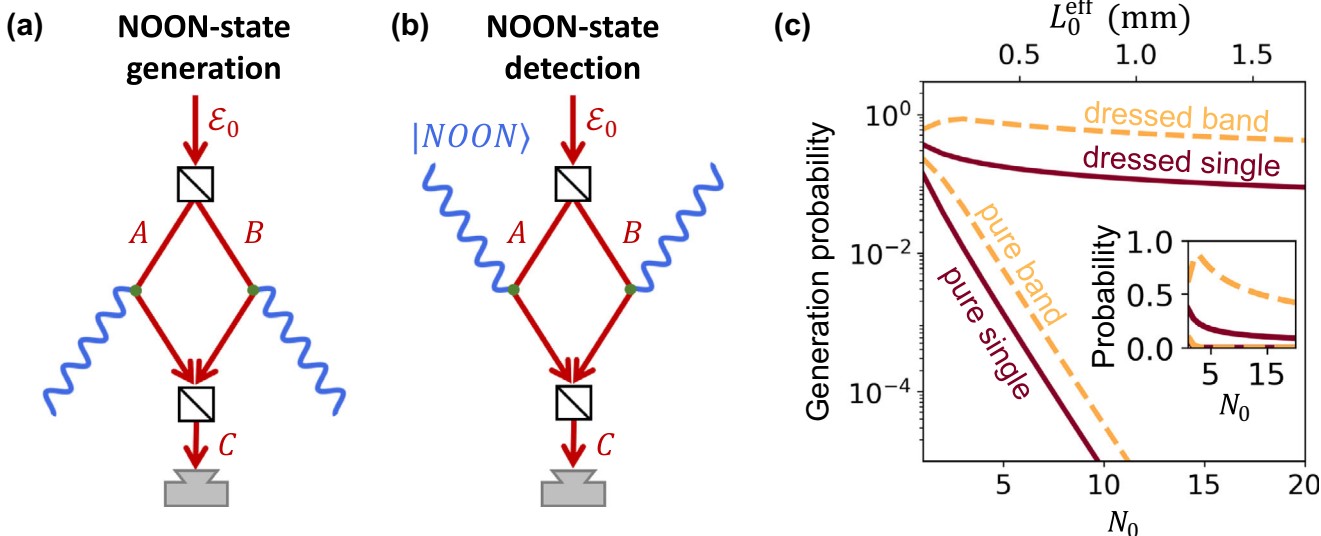

**Fig. 3 | High-NOON state generation. a** QUAFE configuration for generating NOON states consisting of $N_0$ photons in the $n = 0$ waveguide mode and no photons in modes with $n \neq 0$, heralded by the detection of an electron that has lost energy $N_0 \hbar \omega_0$ relative to the incident energy $\mathcal{E}_0$. Electron paths $A$ and $B$ interact with two separate waveguides and are subsequently combined into a single path $C$, where electron detection takes place. **b** QUAFE configuration for detecting NOON states. Using the same electron-path configuration as in (**a**), paths $A$ and $B$ interact with separate waveguided components of the NOON state. The detection probability depends on the interference between the two electron paths and the relative phase $\varphi_\ell$ between the two NOON-state components.

**c** Probability of generating the waveguided NOON state discussed in (**a**) as a function of $N_0$, normalized to the number of transmitted electrons. We compare the probability of producing a pure $N_0$ state (pure single) with the cumulative probability of producing $N_0$, $N_0 \pm 1$, and $N_0 \pm 2$ states (pure band), under the condition that no other waveguide modes (i.e., $n \neq 0$) are excited. We also show the corresponding probabilities regardless of the generation of photons in modes $n \neq 0$ (dressed single and band). The inset shows the same plot on a linear vertical scale. The waveguide parameters are the same as in Fig. 1, with fixed $\mathcal{E}_0 = 200$ keV and $b = 200$ nm, while $L_0^{\text{eff}}$ (upper horizontal scale) is optimized to maximize the probability of the target $N_0$.

Transitions between these states are described by the interaction Hamiltonian[12] $\widehat{\mathcal{H}}_{\text{int}} = -ie\sum_n \omega_n^{-1}\mathbf{v} \cdot \left[\mathbf{E}_n(\mathbf{r})\widehat{a}_n - \mathbf{E}_n^*(\mathbf{r})\widehat{a}_n^\dagger\right]$, where $\mathbf{E}_n(\mathbf{r})$ is the electric field distribution associated with mode $n$. Starting from a given electron–light state $|\psi(t \to -\infty)\rangle$ before the interaction and following a previous analysis[12], the post-interaction state in the interaction picture reads[12]

$$|\psi(t \to \infty)\rangle = \left[\prod_{n=0}^{\infty} e^{i\chi_n} \widehat{\mathcal{S}}_n^{\text{int}}(\beta_n)\right]|\psi(t \to -\infty)\rangle, \quad (2)$$

where

$$\widehat{\mathcal{S}}_n^{\text{int}}(\beta_n) = e^{-\beta_n \widehat{a}_n + \beta_n^* \widehat{a}_n^\dagger} \quad (3)$$

is a displacement operator characterized by a mode-dependent coupling amplitude $\beta_n$ [Eq. (1)], while the phase $\chi_n$ can be absorbed by a global phase associated with each electron path. Under the conditions considered in this work, only a few low-order modes $n$ contribute efficiently to Eq. (2). For a waveguide initially prepared in the photonic vacuum state, the final population of mode $n$ is described by

$$\widehat{\mathcal{S}}_n^{\text{int}}(\beta_n)|0\rangle = e^{-|\beta_n|^2/2} \sum_{N_n=0}^{\infty} \frac{(\beta_n^*)^{N_n}}{\sqrt{N_n!}}|N_n\rangle, \quad (4)$$

which prescribes a Poissonian distribution of $|N_n\rangle$ states with an average number of created photons given by $\langle N_n \rangle = |\beta_n|^2$.

Importantly, each generated photon-number state is entangled with a corresponding change in the electron energy, preventing interference among them. In our scheme, we exploit this property to leverage the quantum properties of photon-number states, enabling

enhanced sensing and metrology. The resulting measurements are then simply averaged over the finite distribution of electron-generated photon numbers (see below).

## Quantum free electrons

The strong interaction between free electrons and waveguided photons can be harnessed to generate NOON states and implement quantum sensing and metrology protocols within the framework of quantum free electronics (QUAFE). This approach involves integrating the electron–photon couplers discussed above with additional components for manipulating light and electrons, such as beam splitters, phase shifters, and electron detectors, as illustrated in Fig. 2. In this work, we combine these elements to develop practical quantum sensing and metrology schemes that achieve super-sensitivity and super-resolution, relying solely on electron current measurements.

While splitters, mixers, and phase shifters for photons are standard in optical setups, analogous elements exist for free electrons in the context of electron microscopy. For example, the electron biprism was developed seven decades ago[45] and can function as a e-beam splitter: a biased wire placed in the path of an electron plane wave deflects electrons on either side into two paths with well-defined angles[46]. A more versatile alternative is provided by transmission gratings, which have been shown to split electron waves into different diffraction orders[37]. By selecting two dominant diffraction directions, they can be used to implement beam splitting and mixing, enabling the realization of a free-electron Mach-Zehnder interferometer[37] as well as the measurement of optical properties in a specimen[38]. In what follows, we consider gratings that split an incident plane wave into two dominant diffraction directions with identical transmission amplitudes. Likewise, we consider gratings that mix two beams into a single one, with each incident beam contributing an identical amplitude. Efficient electron gratings have

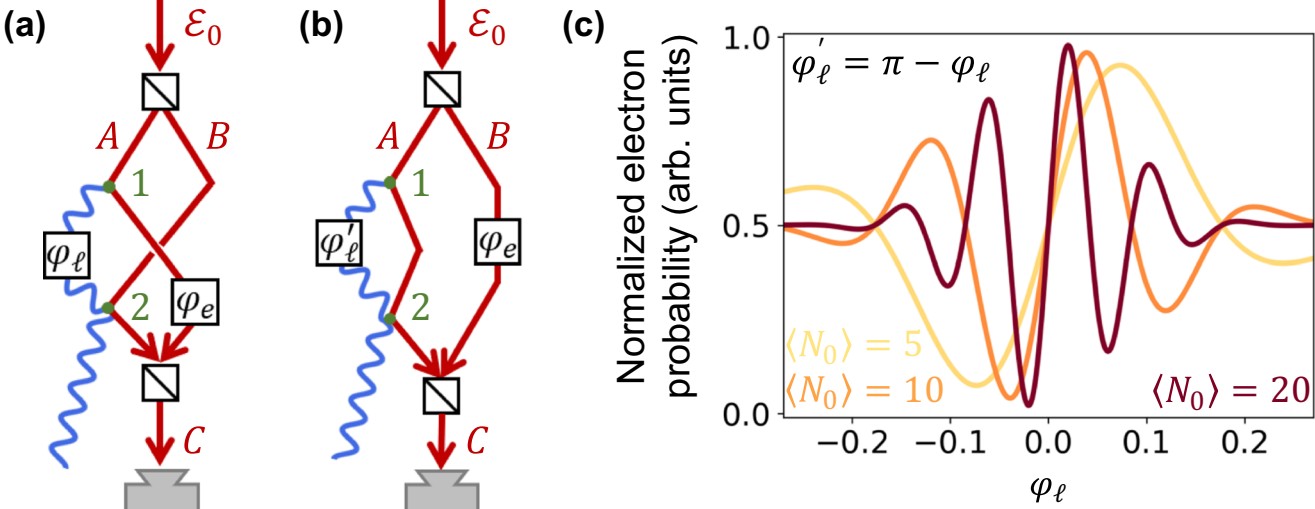

**Fig. 4 | Super-sensitivity and super-resolution through QUAFE. a** Proposed configuration for super-sensitive phase measurements. An electron wave is split into two paths, $A$ and $B$, which are later mixed into a common path $C$. Paths $A$ and $B$ interact with a single optical waveguide. An optical phase $\varphi_\ell$ is introduced between the two electron–light interaction regions. The electron current is measured at the output of path $C$. A controllable electron phase is applied to path $B$ to adjust the relative phase difference between paths $A$ and $B$. Different elements of Fig. 2 are combined in this protocol. The two electron-waveguide interaction points are denoted 1 and 2. **b** Alternative configuration in which only path $A$ interacts with the waveguide, undergoing two separate interactions in different regions. **c** Normalized electron detection probability for the configuration shown in (**a**) as a function of the optical phase $\varphi_\ell$ introduced in waveguide mode $n = 0$. The phases of higher-order modes are assumed to scale proportionally with their respective frequencies. Different curves correspond to selected values of the average photon number in mode $n = 0$ ($\langle N_0 \rangle = |\beta_0|^2$), under the conditions of Fig. 1c–e for 200 keV electrons. The same plot also applies to the configuration in (**b**) by replacing $\varphi_\ell$ with $\pi - \varphi_\ell$.

been demonstrated with amplitudes ~1/4 or higher[37], which introduce an overall correction factor in our results without involving a substantial decrease in electron current.

The electron phase shifter in Fig. 2 operates by multiplying the electron wave function by a phase factor $e^{i\varphi_e}$, while a phase $\varphi_\ell$ applied to an optical waveguide mode $n$ introduces a factor $e^{iN_n\varphi_\ell}$ in the number state with $N_n$ photons. Since we are interested in sensing small optical phases, we assume a mode-dependent phase shift of the form $\varphi_\ell \omega_n / \omega_0$, proportional to the mode frequency (as would result from transmission through a thin dielectric film), with $\varphi_\ell$ representing the phase shift for the fundamental mode $n = 0$.

The operations described in this section constitute the building blocks of QUAFE systems, which can be represented by diagrams that combine the elements shown in Fig. 2, arranged vertically to follow the downward evolution of time and concluding with electron detection (i.e., we focus on protocols based solely on electron measurements). When the electron reaches the detector, the system collapses into a specific electron energy state, or equivalently, a well-defined photonic state due to the aforementioned electron–photon entanglement, assuming monochromatized incident electrons (see above). This property is central to the protocols presented next.

**Efficient generation of high-NOON states**

For tutorial purposes, we first explore the application of QUAFE to generate NOON states with a high photon number, while we present a complete measurement scheme based on electron detection alone in a subsequent section. NOON states can be generated using the configuration shown in Fig. 3a, where the incident electron is split into two paths $A$ and $B$, but now, each of them interacts with a different waveguide. These paths are subsequently recombined into a common path $C$, where electrons are detected with energy resolution much finer than the energy of the generated photons. As explained above for interactions with a single waveguide, the detection of an electron that has lost an energy $\hbar\sum_n N_n \omega_n$ heralds the creation of a photon-number state

$|\{N_n\}\rangle$. In the present configuration, the electron mixer erases which-path information, resulting in the creation of a superposition state $|\{N_n\}, \{0\}\rangle + |\{0\}, \{N_n\}\rangle$, where the two entries in each ket refer to the two different waveguides.

The probability of creating a pure NOON state in the waveguide modes $n = 0$ (i.e., with $N_0 \neq 0$ and $N_n = 0$ for all $n \neq 0$) is given by the product $P_{0,N_0} P_{1,0} P_{2,0} \cdots$, where $P_{n,N} = e^{-P_n} P_n^N / N!$ is a Poissonian distribution of photon-number states created in mode $n$, and $P_n$ is the average photon population. This pure NOON-state generation probability decreases rapidly with $N_0$, as shown in the lowest curve of Fig. 3c. In this plot, the waveguide parameters are the same as in Fig. 1, but the effective interaction length is optimized to maximize $P_{0,N_0}$. In a more practical scenario, the generation of higher-frequency photons ($\omega_{n\neq 0} > \omega_0$), which dress the photonic state of the system, can be tolerated, as they can be filtered out after generation. Then, the NOON-state generation probability is simply $P_{0,N_0}$, which reaches values between 0.1 and 1 for $N_0$ up to 20. For typical beam currents of $10^9$ electrons per second in electron microscopes, this probability translates into a NOON-state generation rate of ~ $10^8$ Hz for $N_0$ ~ $10 - 20$. This high emission probability requires a large effective interaction length $L_{\mathrm{eff}}$ (upper horizontal scale in Fig. 3c), which, for 200 keV electrons and a fixed minimum distance $b = 200$ nm, is achieved at incidence angles in the $\theta_0 = 0.15 - 0.4$ mrad range.

As a complement to electron-based NOON-state generation, a similar scheme could be envisioned for NOON-state detection (Fig. 3b), where two waveguides supporting the NOON state are probed by separate electron paths. However, efficient electron–light interaction requires precise temporal overlap between the photons and the probing electron, demanding challenging temporal synchronization and spectral shaping of the electron wave packet. In addition, the probability for the electron absorbing a large number of photons decreases rapidly (analogously to that in pure NOON-state generation), rendering this approach experimentally impractical. Instead, photon detection and generation by an individual

electron within a QUAFES setup is more advantageous, as we show next, eliminating the need for certified NOON-state generation and surpassing the performance of more traditional optics-based methods.

## Super-sensitivity and super-resolution in QUAFE phase measurements

We present two examples of protocols that enhance sensitivity and resolution in the measurement of an optical phase (Fig. 4a, b). In Fig. 4a, an incident electron is split into two paths ($A$ and $B$), each interacting once with the same waveguide. These two interactions are configured identically, so that the corresponding coupling coefficients $\beta_n$ are equal. The optical phase $\varphi_\ell$ to be measured is introduced in the waveguide modes between the two interaction regions. The electron paths are then recombined into a single path $C$, where the electron current is measured. In the absence of electron–photon coupling, the setup functions as a Mach-Zehnder interferometer, in which we also introduce a controllable phase shift $\varphi_e$ between the two electron paths. Analyzing the electron current $I_e$ using the formalism introduced above, we find

$$I_e \propto 1 + \exp\left[-\sum_n \langle N_n \rangle (1 - \cos\varphi_{\ell n})\right] \times \cos\left(\varphi_e + \sum_n \langle N_n \rangle \sin\varphi_{\ell n}\right), \tag{5}$$

where $\varphi_{\ell n} = \varphi_\ell \omega_n / \omega_0$ (see Methods and Supp. Sec. S1). The electron current oscillates with $\varphi_\ell$ in a manner dependent on the average number of generated photons, as illustrated in Fig. 4c, constructed from Eq. (5) for $\varphi_e = \pi/2$. This current modulation arises from electron scattering at the mixer into undetected channels away from path $C$, depending on the phase difference between electron paths $A$ and $B$, which is governed by the interaction with the waveguide (see Methods). As anticipated, when $\langle N_0 \rangle \gg 1$, the current exhibits sharp oscillations, enabling super-resolution in the measurement of small phase variations. In addition, assuming $\varphi_\ell \ll 1$, Eq. (5) reduces to

$$I_e \propto 1 + \cos\left(\varphi_e + N^{\text{eff}} \varphi_\ell\right), \tag{6}$$

where $N^{\text{eff}} = \sum_n \langle N_n \rangle \omega_n / \omega_0$ represents an effective photon number that takes into account the weighted contributions from all waveguide modes. For the parameters considered in Fig. 4c, we obtain $N^{\text{eff}} \approx 3.3 \times \langle N_0 \rangle$ (see also Supp. Fig. S4), which explains the sharp features observed in the electron current in accordance with Eq. (6).

The sensitivity of this method can be estimated from the maximum slope of the measured current, evaluated as $\partial I_e / \partial \varphi_\ell$ at $\varphi_\ell = 0$, which reveals a linear dependence of current variations on both optical phase variations and the effective photon number: $\Delta I_e \propto N^{\text{eff}} \Delta \varphi_\ell$.

In an alternative configuration, both electron–photon interactions can take place along the same path $A$ (Fig. 4b), leading to analogous results as in the previous configuration, so that Fig. 4c still applies, but with $\varphi_\ell$ substituted by $\pi - \varphi'_\ell$ in the horizontal axis. In practice, after reflection at the waveguide, path $A$ can be electrostatically reflected to be redirected towards a second point of interaction with the waveguide in such a way that the guided modes and the electron are temporally synchronized between the two interaction points. The role of path $B$ in this configuration is to produce interference with the zero-loss peak of path $A$ (i.e., the component of path $A$ that does not produce any photons as a result of its two interactions with the waveguide).

In our QUAFE setups, there are two electron–waveguide interaction points, denoted 1 and 2 in Fig. 4a, b. The difference between the two configurations is that each interaction point involves a different electron path in Fig. 4a, whereas in Fig. 4b, a single path interacts twice

with the waveguide, and the other one does not interact. In these two schemes, the measured phase is introduced in the amplitude of the photons generated at point 1 (i.e., it affects photons generated at point 1 and propagating toward point 2).

Importantly, the proposed configurations require only electron current measurements (i.e., neither photon detection nor electron energy resolution is necessary). This makes the technique robust against variations in incident electron energy, provided the coherent energy spread of individual electrons is smaller than the photon energies involved. This condition is met by low-coherence electron sources, ensuring that different photon-number states created by the electron do not interfere once the electron is traced out.

## Discussion

In summary, we exploit the quantum nature of free electrons to establish, through rigorous theory, optical phase measurements with enhanced sensitivity and resolution, relying solely on the detection of electron currents. Central to this approach is an efficient electron–photon coupling scheme, which we achieve via grazing-angle reflection of energetic electrons from electrically biased one-dimensional waveguides. The electrical bias enforces a minimum electron–waveguide separation of tens of nanometers, preserving coupling to guided modes while avoiding inelastic losses such as electron-hole pairs, which are more localized near the waveguide surface. The bias also produces a smooth, bouncing electron trajectory that enables long effective interaction lengths of hundreds of microns along the waveguide.

As a tutorial example of the potential of this scheme, we show that coherently splitting the electron into a superposition of two paths enables the efficient generation of NOON states with high photon numbers (e.g., $N = 10 - 20$ at $10^8$ Hz rates). While optical NOON-state detection remains a challenge, we circumvent this by formulating an alternative metrology strategy in which individual electrons both generate and probe photonic states, eliminating the need for photon detection. This is a more practical configuration (Fig. 4) in which super-resolution and super-sensitivity are achieved by exploiting the interference between photon emission and absorption by the electron to probe a small optical phase, benefiting from the phase amplification effects associated with large photon-number states. Importantly, these states are mutually incoherent, as each is entangled with distinct final electron energies, so the measurement corresponds to the average over individual photon-number states.

In the analysis of Fig. 4, we consider electrons incident with well-defined angles relative to the waveguide, but in practice, the e-beam has a finite distribution over incident angles that translates into variations in the electron trajectory and different numbers of generated waveguided photons. Importantly, when averaging over incidence angles, this effect does not reduce the achievable sensitivity and resolution within a small range of probed phases (see Supp. Fig. S5). In addition, as discussed above, the repulsive DC field introduced in the waveguide to avoid close electron collisions (see Fig. 1a) could be modulated along the waveguide direction to generate an equal number of photons roughly independent of incidence angle (see Supp. Fig. S3).

Because our proposed schemes rely on the measurement of electron currents, the detrimental effects associated with shot noise and fluctuations in the incident electron current should be addressable by increasing the measurement time. In addition, the QUAFE signal depends on the average number of generated photons, so it is intrinsically robust against photon number fluctuations. In particular, quantum-enhanced sensitivity and resolution are still preserved when including a moderate degree of inelastic attenuation in the propagation of photons from the interaction point 1–2 in Fig. 4a, c, as we show in Supp. Fig. S6.

Our approach leverages existing technologies, including optical waveguides, coherent electron optics, and electron splitters and mixers. In addition, it is robust against incident electron energy

broadening, as the electron–photon coupling is self-regulated by the phase-matching condition, ensuring that successive couplings are automatically synchronized. We conclude that the integration of electron wave splitting, path recombination, and controlled photon coupling constitutes a practical and powerful platform for quantum free-electron technologies, with strong potential to advance quantum sensing and metrology by taking advantage of the unique quantum properties of free electrons.

## Methods

**Electron trajectory and calculation of $L_n^{\text{eff}}$.** We consider the geometry depicted in Fig. 1a, where an electron is reflected from a waveguide by a perpendicular DC electric field $\mathbf{E}_{\text{DC}}$, assumed to be uniform in the region where the guided modes display a non-negligible optical field. The DC field could be generated either by placing two gates far from the waveguide or by slightly doping the waveguide material to make it conductive and biasing it with respect to a distant gate. We consider a minimum electron–waveguide distance $b$, controlled by $E_{\text{DC}}$, the electron energy $\mathcal{E}_0$, and the angle of incidence relative to the waveguide direction $z$. This finite distance is imposed to guarantee that high-energy inelastic losses do not cause electron decoherence, so only waveguide modes play a role (see Supp. Fig. S1). The electron trajectory is determined by $d(mv_x\gamma)/dt = eE_{\text{DC}} + F_{\text{im}}(x)$ (see Supp. Sec. S2), where $v_x$ is the velocity component perpendicular to the waveguide and we introduce $F_{\text{im}}(x) < 0$ as the position-dependent attractive image force, which also varies with electron velocity $v$ and waveguide parameters (see Supp. Sec. S3 for a detailed calculation near the waveguide). Assuming $|v_x| \ll v$ and neglecting the image interaction, we have a parabolic trajectory $x_e(z) \approx b + eE_{\text{DC}}z^2/(2m_e\gamma v^2)$, where $z = 0$ marks the apex and the surface of the waveguide is taken at $x = 0$. However, the image force introduces small but discernible changes in the trajectory (Supp. Fig. S7), so we need to include it in the calculation of the effective interaction length $L_n^{\text{eff}}$ for each mode $n$.

To obtain $L_n^{\text{eff}}$, we integrate the corresponding loss probability per unit path length $dP_n(x)/dz \approx e^{-2x/\lambda_{\perp n}} dP_n(x=0)/dz$ along the trajectory. Notice that $dP_n(x)/dz$ depends exponentially on $x$ to a good approximation, with a characteristic field decay length $\lambda_{\perp n}$ that we fit to numerical BEM calculations (see Supp. Fig. S8). The photon-generation probability of mode $n$ is given by $P_n = \int dz\, dP_n [x_e(z)]/dz = L_n^{\text{eff}} \times dP_n(x=0)/dz$, where the rightmost side defines the effective interaction length $L_n^{\text{eff}}$. After some simple algebra (see Supp. Sec. S4), we find $L_n^{\text{eff}} = e^{-2b/\lambda_{\perp n}}\sqrt{\pi m_e\gamma v^2\lambda_{\perp n}/eE_{\text{DC}}} \times g_{\text{im}}$, where $g_{\text{im}}$ is a correction factor that becomes unity in the absence of image interaction and depends on both $b/\lambda_{\perp n}$ and $F_{\text{im}}(x)/eE_{\text{DC}}$.

**Electromagnetic calculations for the electron energy loss probability and the image force.** We use BEM[41] to compute the loss probability using the frequency-dependent dielectric function of diamond[42]. The image force is obtained from the spectral integral of the frequency-resolved Lorentz force calculated from BEM (see Supp. Sec. S3).

**Derivation of Equation (5).** We are interested in the transmitted electron current in the configuration of Fig. 4 a. After each electron path $A$ and $B$ interacts with a waveguide initially prepared in the photonic vacuum state, the joint electron–light system evolves into the state described by Eq. (4). Considering a single-mode waveguide for simplicity, the electron–light state can be written as $e^{-|\beta|^2/2}\sum_{N=0}^{\infty}(\beta^*)^N(N!)^{-1/2}[e^{i(\varphi_e+N\varphi_\ell)}|A\rangle + |B\rangle] \otimes |N\rangle$, where $|A\rangle$ and $|B\rangle$ represent the two electron paths, the photon-number state $|N\rangle$ is shared by both paths (i.e., there is only one waveguide), and we include the electron and optical phases $\varphi_e$ and $\varphi_\ell$ introduced in path $A$. Note that the optical phase $\varphi_\ell$ needs to be multiplied by the photon number $N$. As discussed above, the mode-dependent phase is taken as $\varphi_\ell\omega_n/\omega_0$. The two paths are now recombined into a single path $C$ (see Fig. 4a). Assuming equal transmission coefficients $A \to C$ and $B \to C$, and extending the model to include all waveguide modes $n$, the amplitude

of the component associated with path $C$ and a photon-number state $|\{N_n\}\rangle$ becomes $\propto [e^{i\phi_{\{N_n\}}} + 1]\prod_n e^{-|\beta_n|^2/2}(\beta_n^*)^{N_n}(N_n!)^{-1/2}$, where $\phi_{\{N_n\}} = \varphi_e + \varphi_\ell\sum_n N_n\omega_n/\omega_0$ is the overall phase accumulated by path $A$ (first term inside the square brackets) relative to path $B$ (second term). Finally, Eq. (5) is readily obtained from the squared modulus of this amplitude after summing over all $\{N_n\}$ combinations. A more detailed derivation of these methods is offered in the Supp. Sec. S1.

## Data availability

The data that support the findings of this study are readily generated from the presented self-contained formalism and are also available from the corresponding author upon request.

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

## Acknowledgements

This work was supported by the European Research Council (Grant No. 101141220-QUEFES), the Spanish MICIU (PID2024-157421NB-I00 and Severo Ochoa CEX2024-001490-S), the European Commission (FET-Proactive 101017720-eBEAM), and the Catalan CERCA Program.

## Author contributions

F.J.G.A. conceived the concept. C.I.V. carried out the numerical calculations. Both authors developed the theory, discussed the results, and wrote the paper.

## Competing interests

The authors declare no competing interests.
