## [Transparent Peer Review file · Nature Communications]

Quantum Sensing and Metrology with Free Electrons

Corresponding Author: Professor Francisco Javier Garcia de Abajo

Version 0:

Reviewer comments:

Reviewer #1

(Remarks to the Author)

The manuscript “Quantum Sensing and Metrology” explores the application possibilities of strong free-electron photon coupling, demonstrating schemes for high-NOON states and electron-current-based, high-sensitivity, high-resolution optical phase measurements. With the experimental realization of a strong electron-photon coupling drawing closer, these ideas present an exciting way forward, connecting electron-photon interaction schemes with quantum optical demands. By focusing fully on quantum metrology and sensing schemes, this manuscript fits well with other current theoretical research on electron-photon coupling for quantum optics, such as GKP states, squeezed states, and entanglement. Similar to previous theoretical work in this field of free-electron-based quantum optics, the schemes require very high coupling strengths between electrons and photons, for which to reach the authors also add a possible pathway. Combining the presented topics and experimental ideas, this manuscript creates a comprehensive and engaging story, which I would like to see published soon. However, to improve the readability and understanding of the paper for a broader community, and to hopefully see an experimental realization soon, I would like to see some points addressed and clarified:

1) It is not sufficiently clear how the electric field strength and the beam trajectory were chosen and why they enable long interaction lengths. Please explain the setup and the connection between the parameters. Is, for example, the electric field set to compensate for the beam convergence angle such that the “outer” region of the electron beam stays at an identical distance to the surface over the whole interaction length? And how do you want to change the experimental setup from the generation of a high-N NOON state (which needs a long interaction length for high coupling) to the double-interaction for optical phase measurement (where the interaction regions should probably not intersect)?

2) The description of the angled electron beam gets slightly mixed up with their theoretical model of a parabolic electron trajectory (see Fig. 1a). A clearer distinction between the experimental setup and the theoretical model, and a more detailed description of the experimental scheme for high interaction (see also Question 1) would be helpful.

3) The authors claim a generation rate of photons (and NOON states) of around 109 (and 108) Hz at “realistic choices of material and electron parameters”. What is the interaction length and coupling constant the authors are using for this estimate? And what beam/waveguide parameters do they assume?

4) In the experimental scheme on high interaction strengths, the authors do not discuss the waveguide sample. Will the sample itself impact the electron beam, leading to a very different trajectory than the presented parabolic one? And will the electric field set between the sample and a gate influence the described experiments, such as the phase measurement or NOON state?

5) The authors use the terms “excitation probability”, “dispersion diagram”, and “loss probability” interchangeably when talking about Fig. 1b. Please clarify their connection/difference, or use a singular term for the same physical quantity.

6) In the QUAFE measurement setup, is the measurable optical phase shift the one accumulating between interaction points A and B? If so, why is the phase-shift dependent on the total photon number N generated via both interactions, and not the photon number from interaction A only? And if one wants to add the measurement to a metrology setup, how would the authors propose transferring the optical phase shift in between the interaction points to have it measured via the electron current?

7) How would electron current fluctuations influence the QUAFE measurement results, and how stable does the current have to be for this setup to work? Do fluctuating photon number states (which you would expect in the electron-induced generation) influence the measurement?

8) Small corrections:

- The equations on $x(z)$ and L_{eff} require a v^2 instead of a v
- Fig. 1e is not exactly mentioned
- Fig. 2 should be mentioned earlier in the text

In total, I think that this manuscript presents some great ideas and shows their experimental feasibility and requirements,

making this paper very interesting especially for the Electron Microscopy and experimental Quantum Optics community. While in the current state, the text is still difficult to understand, I am certain that some minor restructuring and improved explanations will lead to a very good article, suitable for a broader Physics audience.

Reviewer #2

(Remarks to the Author)

I wish to note that I have reviewed this manuscript previously, and many of my comments still stand. Thus, I shall copy some comments from my previous review in unedited form (I have removed or edited comments that the authors have appeared to address).

The manuscript "Quantum Sensing and Metrology with Free Electrons" by Velasco and Garcia de Abajo proposes a scheme using free electrons with photonic structures to generate quantum photonic states such as N00N states. They argue that it should be possible to herald N00N states with "high" probability (between 0.1 and 1), with $N \sim 10$ and rates of around 100 MHz (the rate is set by the repetition rate of current electron sources). The manuscript then proposes to use a similar interaction scheme to measure the accumulated phase of a photonic mode with resolution beyond the standard quantum limit.

The basic concept of the manuscript is sound and is certainly consistent with a now significant body of proposals for generating quantum light with free electrons. The manuscript should be published somewhere, albeit I feel that the manuscript is better suited for a more specialized journal than Nature Communications. There are several areas where I feel the manuscript falls short of exciting a broad audience outside of the field of quantum electron-light interactions.

1. Realism – certainly N00N states are states with impressive metrological properties, and if one could deterministically generate them with electrons in a realistic way which avoids many of the pitfalls of using N00N states, I think this would capture the interest of a lot of people. But this particular proposal seems not robust to a lot of experimental realities. Another issue is that the N00N state is extremely fragile, and is reduced to a mixed state even when one photon is lost. If the photons are generated in a structure like a dielectric waveguide, it seems hard for me to realistically ensure that enough photons would be preserved, especially once the light reaches the waveguide edge, which may be important even for the phase-measurement scheme since the electron-photon entanglement will be diminished. This is true even if the light propagates losslessly over the electron-photon interaction length, as the authors have asserted.

That said, it remains completely unclear that the systems the authors describe as low loss are compatible with the interaction strengths needed to make this proposal realistic. At the end of the day, their interaction strengths rely on the ability to achieve grazing interaction, for which schemes have been proposed by a number of authors, but interaction strengths of the order described in this manuscript remain far from being realized.

Generally, the major "issue" as I see it is that there is no dearth of proposals to herald N00N states – the interesting question in the field is which proposal is realistically possible. And I feel that the analysis in this manuscript does not go far enough to assess the impact of important experimental limitations associated with electron light coupling. I would describe this as one of the main limitations of this work. Ultimately this work assumes highly idealized electron beam dynamics, does not analyze downstream propagation of the generated light, etc.

2. Novelty – along those lines, it is worth point out that there are several theoretical proposals using free electrons to generate various important quantum photonic states such as Fock states, GKP states, magic states, and so on – with even some experimental implementation. See for example work by the groups of Abajo, Kaminer, Fan, and Ropers (Ben Hayun et al. *Sci. Adv.* (2020), Feist et al. (2022), Dahan et al. *PRX* (2023), Rasmussen et al. *Sci. Adv.* (2024), Karnieli et al. *ACS Photonics* (2024)). The current manuscript essentially requires the same elements as described in previous papers (even for "just" creating or detecting Fock states): (a) a substantial probability of generating 1 or more photon by a free electron in a low loss environment [which has never been demonstrated to anywhere near the level required by this paper] (b) electrons to interact with a dielectric structure efficiently for a long length without loss or dephasing [which to my knowledge has also not been demonstrated]. The theoretical framework and tools used in this paper is very conceptually similar to previous works, which in my view, means that this work is partially anticipated by the prior art.

As I said in my previous review, this manuscript should certainly be published somewhere: I think it is an interesting consequence of the current and growing body of work on quantum electron-photon interactions. My main issues are ultimately that the results are somewhat unsurprising in light of that body of work, it does not seem very realistic as written due to a lack of analysis of the main effects that could undermine the proposal, and it does not seem that the existing experimental capabilities are near a level for which this particular proposal becomes urgent.

Reviewer #3

(Remarks to the Author)

In this work, Velasco and García de Abajo tackle an important problem optics in quantum optics, namely, how may one break a standing problem of quantum light and form high-number N00N states. To do so, they propose to use the coupling of free-flying electron with a photonic mode, rather than the standard approach of creating quantum light by mixing classical beams in nonlinear media. The authors provide several examples for how the electron-photon system can be used, if (or

once) it is realized, for phase-sensitive measurements. They suggest that N00N states can be made by post-selecting an equivalent energy loss on an electron spectrometer, or claim that scheme may have a further robustness, as quantum measurement can be the sum of the effect for all the N-states with their entangled electron at an energy $E_0 - N\hbar\omega$, without selecting a particular N-state.

The way I understand this paper is mainly as a concrete proposal to use electron-photon coupling to N00N states, now that we have evidence that electron-photon entanglement survives in the TEM. Thus, it upgrades quantum optics and quantum electron-optics and aims the field towards this useful resource. That aspect is very important, and the path it suggests merits publication in Nature Communications.

With that said about the main claim, there are meaningful issues the authors need to carefully address in the paper:

- The title is too general, giving no information on the paper's content. It is not the first proposal of quantum measurement in an EM, and furthermore, there were quantum measurement experiments, such as the recently demonstrated quantum entanglement by two groups and the correlation microscopy in materials (CLE by LPS). In my view, the paper is about N00N source and N00N-based measurements, and the title should reflect that.
- I couldn't find the results and elaborative analysis for the entanglement-based robustness, namely, having the sensitivity of N00N states without isolating the electron energy. All the figures are on the $\{N_n\}$ states, with a defined N. Although eq. (5) seems to suggest that, I'm not sure it does. For example, in LIGO, a large- $\langle N \rangle$ coherent state, that is a classical (or squeezing-boosted) state is used to improve sensitivity. The authors should explain how the $\langle N \rangle$ scaling is proof of quantum enhancement, which is typically assessed with respect to the noise.
- Minor issues - what does "theoretically demonstrate" in the conclusion mean? Seems like a "propose" is a better term.

Version 1:

Reviewer comments:

Reviewer #1

(Remarks to the Author)

The authors have addressed all issues raised, improving the clarity of their theory. Especially the enhanced first section on how to achieve strong electron-light coupling is now better understandable, providing a basis for the rest of the work. I also appreciate the more detailed supplementary information, which gives further information and proof without making the main text too equation-heavy. I think the current version is suitable for publication.

Reviewer #2

(Remarks to the Author)

I have reviewed the reply to my comments and those of the other Reviewers. While I still feel that this work remains speculative, and requires advances that seem far in the future relative to the existing art, the manuscript is sound, and could be published.

made.

Barcelona, October 20th 2025

We thank the Reviewers for their constructive reviews on our paper. Next, we present a detailed point-by-point response, along with a description of changes made to the manuscript (**highlighted in yellow**). We also provide a **blue-lined** version of the modified paper.

Reviewer #1

Reviewer #1. The manuscript “Quantum Sensing and Metrology” explores the application possibilities of strong free-electron photon coupling, demonstrating schemes for high-NOON states and electron-current-based, high-sensitivity, high-resolution optical phase measurements. With the experimental realization of a strong electron-photon coupling drawing closer, these ideas present an exciting way forward, connecting electron-photon interaction schemes with quantum optical demands. By focusing fully on quantum metrology and sensing schemes, this manuscript fits well with other current theoretical research on electron-photon coupling for quantum optics, such as GKP states, squeezed states, and entanglement. Similar to previous theoretical work in this field of free-electron-based quantum optics, the schemes require very high coupling strengths between electrons and photons, for which to reach the authors also add a possible pathway. Combining the presented topics and experimental ideas, this manuscript creates a comprehensive and engaging story, which I would like to see published soon. However, to improve the readability and understanding of the paper for a broader community, and to hopefully see an experimental realization soon, I would like to see some points addressed and clarified:

Our reply. We thank the Reviewer for the appraisal of our work and for the insightful comments, which we have addressed as explained in the point-by-point reply below.

Reviewer #1. 1) It is not sufficiently clear how the electric field strength and the beam trajectory were chosen and why they enable long interaction lengths. Please explain the setup and the connection between the parameters. Is, for example, the electric field set to compensate for the beam convergence angle such that the “outer” region of the electron beam stays at an identical distance to the surface over the whole interaction length? And how do you want to change the experimental setup from the generation of a high-N NOON state (which needs a long interaction length for high coupling) to the double-interaction for optical phase measurement (where the interaction regions should probably not intersect)?

Our reply. We have explained the choice of DC field and trajectory with further discussion in the revised “Introduction” and “Efficient electron–photon coupler” sections, and also prepared a new Supp. Sec. 2 with details of the calculation of the trajectory. Indeed, the field is chosen to repel the electron and prevent it from reaching the waveguide, thus avoiding decoherence caused by close interactions with inelastic modes such as electron-hole pairs in the material. We also explain that, as the electron trajectory depends on the angle of incidence, these angles are associated with different positions of arrival of the electron along the waveguide, so we can vary the DC field slowly along the waveguide and exploit this possibility to compensate for a small divergence angle $\Delta\theta$ (finite distribution over incidence angles) of the incident beam around a central incidence angle θ_0 , such that the distance of closest approach to the waveguide is made roughly independent of angle within that small angular range (see new Supp. Fig. S3). In addition, we have added a new Supp. Fig. S5 with a plot of the average of electron probabilities over a finite range of electron beam divergence angles, showing that the result is robust against this parameter (i.e., such an average preserves super-resolution and super-sensitivity). For the double electron-waveguide interaction in Fig. 4b, we have included an explanation of how this could be implemented in practice.

Reviewer #1. 2) The description of the angled electron beam gets slightly mixed up with their theoretical model of a parabolic electron trajectory (see Fig.1a). A clearer distinction between the experimental setup and the theoretical model, and a more detailed description of the experimental scheme for high interaction (see also Question 1) would be helpful.

Our reply. We present calculations for uniform DC fields that produce reflection of the electron at distances of tens of nanometers from the surface. This reflection away from the surface is introduced not only to extend the portion of the trajectory in which the electron interacts with the waveguide but also to avoid close collisions with the material that would lead to incoherent losses suffered by the electron (e.g., by excitation of electron-hole pairs across the band gap). We have emphasized this point in the revised “Efficient electron–photon coupler” subsection. Incidentally, we have corrected two typos that mixed incidence and divergence angles, so we hope the text is clearer now.

Reviewer #1. 3) The authors claim a generation rate of photons (and NOON states) of around 109 (and 108) Hz at “realistic choices of material and electron parameters”. What is the interaction length and coupling constant the authors are using for this estimate? And what beam/waveguide parameters do they assume?

Our reply. We thank the Reviewer for this comment, as we realize that this information (previously embedded in the calculations) was not shown explicitly. To facilitate the reading of physical parameters, we have added an upper horizontal scale to Fig. 3c showing the effective interaction length. In addition, we have revised

the main text to make this point clearer and to ensure that the waveguide parameters are kept constant throughout the paper (a diamond waveguide with a width of 600 nm and a height of 800 nm). Also, we have updated Fig. 1c-e with slight changes in the choice of parameters.

Reviewer #1. 4) In the experimental scheme on high interaction strengths, the authors do not discuss the waveguide sample. Will the sample itself impact the electron beam, leading to a very different trajectory than the presented parabolic one? And will the electric field set between the sample and a gate influence the described experiments, such as the phase measurement or NOON state?

Our reply. We expect a small effect of the material, but we have corrected the calculation, as explained below, to produce an accurate description of the electron trajectory. Let us first clarify that inelastic losses other than those associated with the excitation of guided modes are prevented by maintaining a minimum electron-waveguide distance above 200 nm, and in addition, photon generation does not change the electron velocity substantially (nonrecoil approximation). A more important effect relates to the conservative image interaction, which decreases when considering relativistic electrons compared to a static charge in front of a material, and also because of the finite lateral size of the waveguide. The effect of such an image interaction is now included in the revised text and calculated in the new Supp. Sec. S2, with details of the derivation of the image force in the new Supp. Sec. S3. This produces small, but discernible corrections to the trajectory and the effective interaction length, as discussed in the new Supp. Sec. S4 (see also new Supp. Fig. S7).

Reviewer #1. 5) The authors use the terms “excitation probability”, “dispersion diagram”, and “loss probability” interchangeably when talking about Fig. 1b. Please clarify their connection/difference, or use a singular term for the same physical quantity.

Our reply. The generation of a waveguide photon comes at the expense of electron energy. Every photon generation event is thus associated with a loss of energy by the electron (i.e., the energy-loss probability within the energy gap of the material can be interpreted as a photon-generation probability). In addition, the dispersion diagram is constructed from the electron energy-loss probability plotted as a function of energy loss and momentum transfer. We have clarified these points in the revised text and also changed “excitation” to “photon-generation”.

Reviewer #1. 6) In the QUAPE measurement setup, is the measurable optical phase shift the one accumulating between interaction points A and B? If so, why is the phase-shift dependent on the total photon number N generated via both interactions, and not the photon number from interaction A only? And if one wants to add the measurement to a metrology setup, how would the authors propose

transferring the optical phase shift in between the interaction points to have it measured via the electron current?

Our reply. In both schemes, photons are generated at two points, denoted 1 and 2 in the revised Fig. 4. Then, a phase shift is introduced only for the amplitude of photons that have been generated at point 1. We have clarified this aspect at the end of the revised QUAPE section. In addition, we provide details of the formalism in the new Supp. Sec. S1, where the role of this phase is more clearly identified.

Reviewer #1. 7) How would electron current fluctuations influence the QUAPE measurement results, and how stable does the current have to be for this setup to work? Do fluctuating photon number states (which you would expect in the electron-induced generation) influence the measurement?

Our reply. We are proposing schemes that rely on the measurement of an electron current. This can be sensitive to shot noise as well as fluctuations in the incident electron-beam current. However, we understand that this situation is similar to regular EELS or CL experiments, where a reasonably stable current and a sufficient number of counts are needed. In addition, the QUAPE signal depends on $\langle N \rangle$ (the average number of generated photons), and thus, it should be rather insensitive to fluctuations in photon numbers. We have clarified this point in the revised Discussion section. In addition, the proposed method is relatively immune to inelastic attenuation in the waveguide photons, as demonstrated in the new Supp. Sec. S1 and Supp. Fig. S6.

Reviewer #1. 8) Small corrections:

- The equations on $x(z)$ and L_{eff} require a v^2 instead of a v
- Fig. 1e is not exactly mentioned
- Fig. 2 should be mentioned earlier in the text

Our reply. We thank the Reviewer for pointing out these issues, which are now fixed in the revised manuscript.

Reviewer #1. In total, I think that this manuscript presents some great ideas and shows their experimental feasibility and requirements, making this paper very interesting especially for the Electron Microscopy and experimental Quantum Optics community. While in the current state, the text is still difficult to understand, I am certain that some minor restructuring and improved explanations will lead to a very good article, suitable for a broader Physics audience.

Our reply. We thank the Reviewer for the constructive criticism, in response to which we have substantially revised the manuscript. We believe that the new version is more pedagogical, including a detailed description of practical design parameters and a clearer and comprehensive derivation of the formalism.

Reviewer #2

Reviewer #2. I wish to note that I have reviewed this manuscript previously, and many of my comments still stand. Thus, I shall copy some comments from my previous review in unedited form (I have removed or edited comments that the authors have appeared to address).

Our reply. We thank the Reviewer for examining our paper again. Indeed, we previously submitted our manuscript to another journal, and we believe that we addressed all the concerns of the Reviewers from that journal. Below, we address in more detail the points raised by the Reviewer in our point-by-point fashion.

Reviewer #2. The manuscript “Quantum Sensing and Metrology with Free Electrons” by Velasco and Garcia de Abajo proposes a scheme using free electrons with photonic structures to generate quantum photonic states such as NOON states. They argue that it should be possible to herald NOON states with “high” probability (between 0.1 and 1), with $N \sim 10$ and rates of around 100 MHz (the rate is set by the repetition rate of current electron sources). The manuscript then proposes to use a similar interaction scheme to measure the accumulated phase of a photonic mode with resolution beyond the standard quantum limit.

The basic concept of the manuscript is sound and is certainly consistent with a now significant body of proposals for generating quantum light with free electrons. The manuscript should be published somewhere, albeit I feel that the manuscript is better suited for a more specialized journal than Nature Communications. There are several areas where I feel the manuscript falls short of exciting a broad audience outside of the field of quantum electron-light interactions.

Our reply. We would like to clarify that the free-electron-based generation of NOON states is presented as a tutorial section that should prepare the reader to better understand the rest of the paper. However, our main results are in the QUAPE configuration of Fig. 4, where we present a new method to perform a substantially improved quantum sensing and metrology of a photonic phase by only measuring electron currents. We also understand that our scheme is combining free electrons with quantum sensing and metrology in an innovative way and with a dramatic gain in sensitivity and resolution that should make the paper interesting to the broad quantum physics community as well as to the free-electron physics community. **We have emphasized these aspects in the revised manuscript.**

Reviewer #2. 1. Realism – certainly NOON states are states with impressive metrological properties, and if one could deterministically generate them with electrons in a realistic way which avoids many of the pitfalls of using NOON states,

I think this would capture the interest of a lot of people. But this particular proposal seems not robust to a lot of experimental realities. Another issue is that the NOON state is extremely fragile, and is reduced to a mixed state even when one photon is lost. If the photons are generated in a structure like a dielectric waveguide, it seems hard for me to realistically ensure that enough photons would be preserved, especially once the light reaches the waveguide edge, which may be important even for the phase-measurement scheme since the electron-photon entanglement will be diminished. This is true even if the light propagates losslessly over the electron-photon interaction length, as the authors have asserted.

Our reply. We again note that we discuss the generation of NOON states as a tutorial way of introducing QUAPE (see main results in Fig. 4). We do not understand what the Reviewer refers to when commenting that “this particular proposal seems not robust to a lot of experimental realities”. In our view, our QUAPE approach is robust, as supported by detailed considerations of design parameters and theory. For clarity, we note that we do not use NOON states for sensing and metrology. Instead, QUAPE relies on photon interferences between photon amplitudes generated at different electron-waveguide interaction points. We believe that **this aspect is now made clearer in the revised manuscript**. In addition, QUAPE is robust with respect to photon attenuation (see the **new Supp. Sec. S1**), and we do not rely on photon measurements. Also note that we avoid the detection of NOON states (and also the dramatic effects produced by optical losses in an optical approach to quantum sensing), which remains challenging for large N ; instead, we rely on the measurement of electron currents using conventional electron detectors and without requiring electron spectroscopy, rendering the method more tolerant to photon losses than all-quantum-optics methods (see **new Supp. Fig. S6**).

Reviewer #2. That said, it remains completely unclear that the systems the authors describe as low loss are compatible with the interaction strengths needed to make this proposal realistic. At the end of the day, their interaction strengths rely on the ability to achieve grazing interaction, for which schemes have been proposed by a number of authors, but interaction strengths of the order described in this manuscript remain far from being realized.

Our reply. We emphasize in the revised manuscript that a large photon generation probability (normalized per electron) is achieved thanks to a new essential ingredient in our proposed configuration (Fig. 1): the presence of a repulsive DC electric field. This also has the effect of preventing close electron-waveguide collisions that would cause decoherence in the electron. **We have amended the manuscript to make this point clearer. Note that we specify a set of realistic parameters alongside rigorous calculations based on them**, which lead to a much higher photon generation probability than in previous works within this field.

Reviewer #2. Generally, the major “issue” as I see it is that there is no dearth of proposals to herald NOON states – the interesting question in the field is which is proposal is realistically possible. And I feel that the analysis in this manuscript does not go far enough to assess the impact of important experimental limitations associated with electron light coupling. I would describe this as one of the main limitations of this work. Ultimately this work assumes highly idealized electron beam dynamics, does not analyze downstream propagation of the generated light, etc.

Our reply. In the revised manuscript, we provide a **new Supp. Sec. S1 with a comprehensive derivation of the theory** (previously only succinctly sketched in Methods), **as well as details on design parameters**, which show how our proposed methods are feasible using currently existing technology. In particular, we devote a substantial part of the manuscript to describing details associated with beam propagation (e.g., beam divergence), so we understand that our results go well beyond an idealized dynamics model.

Reviewer #2. 2. Novelty – along those lines, it is worth point out that there are several theoretical proposals using free electrons to generate various important quantum photonic states such as Fock states, GKP states, magic states, and so on – with even some experimental implementation. See for example work by the groups of Abajo, Kaminer, Fan, and Ropers (Ben Hayun et al. Sci. Adv. (2020), Feist et al. (2022), Dahan et al. PRX (2023), Rasmussen et al. Sci. Adv. (2024), Karnieli et al. ACS Photonics (2024)). The current manuscript essentially requires the same elements as described in previous papers (even for “just” creating or detecting Fock states): (a) a substantial probability of generating 1 or more photon by a free electron in a low loss environment [which has never been demonstrated to anywhere near the level required by this paper] (b) electrons to interact with a dielectric structure efficiently for a long length without loss or dephasing [which to my knowledge has also not been demonstrated]. The theoretical framework and tools used in this paper is very conceptually similar to previous works, which in my view, means that this work is partially anticipated by the prior art.

Our reply. We would like to stress that our formalism introduces additional elements that were not present in the publications indicated by the Reviewer (most of which were already cited in the previous version of the manuscript). In particular, we propose a realistic design to achieve an electron-waveguide interaction over a large length, reaching a large number of photons created per electron while preserving the coherence of the electron. **We have emphasized these aspects in the revised manuscript.** This is a central part of our manuscript to which we devote Fig. 1 and the bulk of the Results section because our proposed sensing and metrology methods depend critically on it. In our view, this is a nontrivial development that contains strong elements of conceptual novelty. In addition, our framework differs substantially from those previous works because we are invoking lateral splitting of

the electron and an innovative way of leveraging electron-photon interactions that was not considered previously.

Reviewer #2. As I said in my previous review, this manuscript should certainly be published somewhere: I think it is an interesting consequence of the current and growing body of work on quantum electron-photon interactions. My main issues are ultimately that the results are somewhat unsurprising in light of that body of work, it does not seem very realistic as written due to a lack of analysis of the main effects that could undermine the proposal, and it does not seem that the existing experimental capabilities are near a level for which this particular proposal becomes urgent.

Our reply. We respectfully disagree with the Reviewer and find the results surprising because we are presenting what we believe are highly nontrivial methods to determine optical phases with a sensitivity and resolution comparable to all-optics methods using up to $N = 20$ photon-number states, which is well beyond the current experimental limits in quantum optics. Instead, our proposal is based on existing technological capabilities, as shown by the added details on parameters for the different elements involved in our designs. Finally, we have made an effort to make the theory more tutorial through the addition of new, detailed sections in the supplementary information.

Reviewer #3

Reviewer #3. In this work, Velasco and García de Abajo tackle an important problem optics in quantum optics, namely, how may one break a standing problem of quantum light and form high-number N00N states. To do so, they propose to use the coupling of free-flying electron with a photonic mode, rather than the standard approach of creating quantum light by mixing classical beams in nonlinear media. The authors provide several examples for how the electron-photon system can be used, if (or once) it is realized, for phase-sensitive measurements. They suggest that N00N states can be made by post-selecting an equivalent energy loss on an electron spectrometer, or claim that scheme may have a further robustness, as quantum measurement can be the sum of the effect for all the N-states with their entangled electron at an energy $E_0 - N\hbar\omega$, without selecting a particular N-state.

The way I understand this paper is mainly as a concrete proposal to use electron-photon coupling to N00N states, now that we have evidence that electron-photon entanglement survives in the TEM. Thus, it upgrades quantum optics and quantum electron-optics and aims the field towards this useful resource. That aspect is very important, and the path it suggests merits publication in Nature Communications. With that said about the main claim, there are meaningful issues the authors need to carefully address in the paper:

Our reply. We thank the Reviewer for the appraisal of our work and for the insightful comments, which we have addressed as explained in the point-by-point reply below. We would like to clarify that the free-electron-based generation of NOON states is presented as a tutorial section that should prepare the reader to understand the rest of the paper. Then, our main results are in the QUAFE configuration of Fig. 4, where we present a new method to perform substantially improved quantum sensing and metrology of a photonic phase by only measuring electron currents.

Reviewer #3. - The title is too general, giving no information on the paper's content. It is not the first proposal of quantum measurement in an EM, and furthermore, there were quantum measurement experiments, such as the recently demonstrated quantum entanglement by two groups and the correlation microscopy in materials (CLE by LPS). In my view, the paper is about NOON source and NOON-based measurements, and the title should reflect that.

Our reply. As explained above, the main result of the paper is not about NOON states: we have introduced this part just as a tutorial section in preparation for the rest of the results in the paper. In reality, the QUAFE configurations in Fig. 4 do not even involve NOON states, as we only have one waveguide. Furthermore, these configurations do not require measuring any light at all. We therefore believe that the title adapts well to a new way of performing sensing and metrology via the use of free electrons, and respectfully request that we maintain it as is.

Reviewer #3. - I couldn't find the results and elaborative analysis for the entanglement-based robustness, namely, having the sensitivity of NOON states without isolating the electron energy. All the figures are on the $\{N_n\}$ states, with a defined N . Although eq. (5) seems to suggest that, I'm not sure it does. For example, in LIGO, a large- $\langle N \rangle$ coherent state, that is a classical (or squeezing-boosted) state is used to improve sensitivity. The authors should explain how the $\langle N \rangle$ scaling is proof of quantum enhancement, which is typically assessed with respect to the noise.

Our reply. We hope that the comprehensive description of the theory presented in the new Supp. Sec. S1 will support the validity of Eq. (5). The result for the variation of the total electron current (without energy resolution in the electron) depends on $\langle N \rangle$ and, thus, on the average number of created photons. In particular, Fig. 4c does not correspond to a fixed choice of $\{N_n\}$, but rather to the final results of the current measurement, which depends on $\langle N \rangle$. It is then immune to fluctuations in photon number and also tolerant to a certain amount of inelastic photon attenuation, as shown in the new Supp. Sec. S1. The quantum enhancement in the determination of optical phases is presented in Eq. (6), which is now displayed to emphasize this result.

Reviewer #3. - Minor issues - what does “theoretically demonstrate” in the conclusion mean? Seems like a “propose” is a better term.

Our reply. We have change “theoretically demonstrate” to “establish through rigorous theory”, which we believe is delivers a clearer message.

We look forward to hearing from you about the status of our paper.

Sincerely yours,

Prof. Javier García de Abajo
ICFO – The Institute of Photonic Sciences